# Effects of the Processing Parameters of Friction Stir Processing on the Microstructure, Hardness and Tribological Properties of SnSbCu Bearing Alloy

**DOI:** 10.3390/ma13245826

**Published:** 2020-12-21

**Authors:** Beata Leszczyńska-Madej, Marcin Madej, Joanna Hrabia-Wiśnios, Aleksandra Węglowska

**Affiliations:** 1Faculty of Non-Ferrous Metals, AGH University of Science and Technology, Mickiewicza 30 Avenue, 30-059 Krakow, Poland; hrabia@agh.edu.pl; 2Faculty of Metals Engineering and Industrial Computer Science, AGH University of Science and Technology, Mickiewicza 30 Avenue, 30-059 Krakow, Poland; mmadej@agh.edu.pl; 3Łukasiewicz Research Network—Welding Institute, Błogosławionego Czesława 16-18 Avenue, 44-100 Gliwice, Poland; Aleksandra.Weglowska@is.gliwice.pl

**Keywords:** SnSbCu alloy, (FSP) friction stir processing, microstructure, hardness, tribological properties

## Abstract

In the study, the friction stir processing (FSP) method was used to modify the surface layer of a tin-based bearing alloy. The modification was aimed at extending the service life of bearings by improving their tribological properties. The results of investigations of the microstructure, hardness and tribological properties of the SnSbCu bearing alloy after FSP using various rotational speeds of the tool—280, 355, 450 and 560 RPM—and the constant traverse speed of 355 mm/min are presented. Particular attention was paid to the possibility of changing the morphology of the precipitates present in the alloy, and to the impact of this parameter on improvement of the tribological properties. The research carried out in this paper covered investigations of the microstructure using light and scanning electron microscopy (SEM) along with analysis of the chemical composition in micro-areas and Brinell hardness tests. Additionally, the sizes of the SnSb and CuSn precipitates present in the microstructure before and after the modification process were determined, as were the tribological properties under technically dry friction conditions and lubrication with TU 32 oil. It was proven that using friction stir processing favors refinement of the microstructure and improves the tribological properties of the analyzed alloy.

## 1. Introduction

Two of the most important properties which sliding bearings should have are wear resistance and a low coefficient of friction. The increasing demands on the operational properties of manufactured elements present even more challenges for surface engineering. One of the promising methods of microstructure modification in surface layers is a new method—FSP—friction stir processing. This method was developed by Mishra et al. [1] based on the basic principles of friction stir welding technology (FSW) [2]. However, the FSP method, unlike FSW, is used to modify the microstructure of the materials, and not to join metal elements. The process consists of heating and plasticizing the material as a result of the friction of a tool equipped with a shoulder and a pin plunged into the material and moving along the modified surface of the element. The movement of the tool causes heating, intense stirring and densification of the deformed material. The research presented in the work of M St. Węglowski [3] and V. Sharma et al. [4] shows that the microstructure in the process zone depends on the parameters of the process (rotational speed of the tool, traverse speed, axial force), the type of material being processed and the shape of the tool.

Compared to traditional metal forming techniques, FSP has some advantages that are worth emphasizing. First, the FSP modification process requires no complicated tooling and can be carried out with simple manually or numerically-controlled milling machines. Other techniques, such as heat treatment and surface texturing, are relatively time-consuming and complex, resulting in a higher cost, while FSP technology produces the desired results in just one operation. Second, the microstructure and mechanical properties of the modified zone can be precisely controlled by optimizing the tool design, FSP parameters and active cooling/heating. Third, FSP is considered a “green” technology that is environmentally-friendly, as it does not release welding gases, create noise, create slag or create magnetic fields. Moreover, the heat (thermal energy) required to modify the microstructure comes from friction and plastic deformations, not from external sources [1,3,4]. Fourth, FSP does not change the shape or size of the processed elements, which was described in the work of M. Ebrahimi et al. [5].

In the relatively short time since its invention, FSP technology has found numerous applications. Initially, this process was used to increase the susceptibility to plastic deformation in aluminum and magnesium alloys; the results can be found, for example, in the work of I. Charit et al. [6], M. St. Węglowski et al. [7] and M. Kopyściański et al. [8]. The use of this method to produce SiC reinforced composite layers on an aluminum surface in 2003 [9] contributed to the development of various components based on copper [10], titanium [11] and steel [12] alloys in the following years. FSP technology is also effective in homogenizing aluminum alloys obtained by powder metallurgy [13], and modifying the microstructure of metal matrix composites [14]. Moreover, FSP effectively eliminates casting defects, which is described in the work of K. Nakata et al. [15], Z.Y.Ma et al. [16] and N. Sun et al. [17]. The novel applications of solid-state technology based on FSP principles are appraised for use in the additive manufacturing (AM) of different materials (metals, alloys and MMC) as variations of so-called friction stir additive manufacturing (FSAM). A detailed description of this technology is presented in the work of F. Khodabakhshia [18].

The literature concerning tin-based Babbits contains information on modifying the microstructure/surface of these alloys in order to improve the mechanical properties, in particular the tribological properties.

Y. Ni et al. [19] proved that using the process of laser remelting of the tin Babbit surface promotes homogenization and refinement of the microstructure and improvement of the tribological properties, including reduction of the wear rate of this alloy. In the work of B.A. Potekhin et al. [20], various casting methods were used, including the “turbulent” casting method developed by the authors of the cited work, which made it possible to obtain a tin bearing alloy with the most beneficial microstructure with globular precipitates of intermetallic phases and the most favorable tribological properties. To improve the tribological properties of the material, H. Zhang et al. [21] used surface texturing. Leszczyńska-Madej et al. [22] proved that changes in the morphology and size of the intermetallic phases obtained thanks to Babbit heat treatment favor improvements of the properties. The latest works on tin Babbits are focused on the production of composites based on these alloys [23] or composites infiltrated with a tin Babbitt alloy [24].

In the literature on bearing alloys, there is no information on attempts to implement FSP to change/modify the phase morphology and possibly improve the tribological properties of these alloys. Earlier works of the authors on bearing alloys and the modifications of their microstructure, and works by other authors, confirmed that even slight changes in the morphology and size present in the microstructure phases can have positive effects on the properties of these alloys.

The aim of the research presented in the paper was to determine the applicability of the FSP method to modify bearing alloys—in particular, to change the morphologies of CuSn and SnSb phases present in the microstructure and their impacts on improvement of the tribological properties.

## 2. Materials and Methods

The research material was the tin-based foundry bearing alloy SnSb11Cu6. The alloy was cast into cast-iron molds, then cooled in ambient air. The chemical composition of the investigated alloy is presented in Table 1.

The FSP studies were performed on a welding stand, built on the basis of an FYF32JU2 vertical milling machine. The alloy was subjected to friction stir processing (FSP). The process was performed using a Triflute pin, at various rotational speeds of the tool—280, 355, 450 and 560 RPM—and a constant traverse speed of 355 mm/min. FSP is illustrated schematically in Figure 1, which also shows the tool used in the process, and photos of samples after the modification process are shown in Figure 2. The surface defects shown in Figure 2 for the pin speeds of 450 and 560 RPM resulted from improper pin recess or from overly high pin rotational speed relative to the traverse speed. This chipping was located only on the surface layer, and it is easy to remove, which allows the exposure of a material with very good properties compared to the reference material and materials modified at lower rotational speeds.

Investigations were carried out on the starting material after casting and after FSP. Examination of the microstructure of the samples was carried out using light (Olympus GX 51 microscope, Tokyo, Japan) and scanning electron microscopy, along with analysis of the chemical composition in energy dispersive X-ray analysis (EDS) micro-areas (Hitachi SU 70 microscope, Tokyo, Japan).

Particle size analysis was performed using the Image FIJI program [25] that enables operations related to image processing and basic particle measurements and analysis to be performed. The particle analysis was performed based on SEM micrographs. Additionally, Brinell hardness measurements were performed using an Innovatest hardness tester (INNOVATEST Europe BV, Maastricht, The Netherlands); a cemented carbide ball with a diameter of 2.5 mm and a load of 31.25 kG was used. The average hardness values and standard deviations were determined.

The abrasive wear resistance test was carried out on a T-05 block-on-ring tester (ITEE, Radom, Poland). The measurement was carried out at ambient temperature, with translational motion in sliding dry and lubricated contact, using TU 32 oil, the specifications of which are in accordance with DIN 51,515 part 1 and ISO 8068. A diagram and detailed description of the T-05 tester are presented in paper [22]. The sample (20 mm × 4mm × 4 mm) was mounted in a sample holder equipped with a hemispherical insert ensuring proper contact between the sample and a rotating ring (counter-specimen). The wear surface of the sample was perpendicular to the pressing direction. A double lever system input the load, pressing the sample to the ring with the accuracy of ±1%. The ring rotated at a constant speed. The friction surface changes during the test: initially there is linear contact, which changes to surface contact with the duration of the test and advancing wear. In any case, the roughness of each sample (initial and after FSP) measured by confocal microscopy was below 1 µm. In order to verify the behavior of the studied alloy in conditions simulating the multiple start-ups and multiple stops of a turbine, a “start–stop” test was conducted over the distance of 10,000 m. The test consisted of carrying out 10 consecutive friction cycles of 1000 m each with TU 32 oil as the lubricant. Table 2 summarizes the applied parameters of the friction process. The test was performed for three samples of each variant; the determined values of the coefficient of friction and weight loss are the average values of these three tests. The tester used enabled tests to be carried out in accordance with the methods specified in the ASTM D 2714, D 3704, D 2981 and G 77 standards.

During the tribological test, friction force F (required to determine the coefficient of friction) was continuously recorded, and the weight loss was determined based on the difference in weight before and after the process. After the tests, the surfaces of the samples obtained in the tribological contact were observed using a scanning electron microscope.

## 3. Results and Discussion

### 3.1. Microstructure Characterization

The investigated alloy is characterized by a multiphase microstructure, consisting of large diamond-shaped hard SnSb phases and numerous needle-shaped and nearly globular-shaped CuSn phase precipitates (Figure 3). The matrix of the alloy is a solution of antimony and copper in tin [22,26].

Photos of the microstructure of the SnSb11Cu6 alloy after FSP, taken using a light microscope, are shown in Figure 4. The results indicate a change in the morphology of the phases present in the investigated alloy. Both the SnSb and CuSn precipitates were significantly refined, and the shape changed to nearly globular (Figure 4). It was found that the needle-shaped form of the CuSn phase was eliminated in favor of the globular shape. In addition, an increase in the rotational speed of the tool favored more regular distribution of the CuSn phase in relation to the distribution in the starting material. Strong refinement of the tin-rich matrix was also observed in the FSP-modified area. FSP generates a significant increase in temperature due to friction forces, intense plastic deformation and material flow caused by the movement of the tool, thereby promoting dynamic recrystallization in the stir zone. In this case, fine and equiaxed recrystallized tin-rich matrix grains of 20–50 µm were formed in the stir zone.

Moreover, no discontinuities or pores were observed in the microstructure, nor was there any distinct characteristic nucleus. Only a change in the structure or shape of the stir zone with an increase in the rotational speed of the tool was noticed (Figure 4). At lower rotational speeds (280 and 355 RPM), material stacking in the form of so-called “onion rings” was visible. This was due to a change in the amount of heat introduced to the stir zone and its temperature. Increasing the rotational speed of the tool results in a higher temperature of the material just below the surface of the tool, and thus it is able to withstand lower loads while becoming more plastic; it is “sheared” faster, and the impact depth of the shoulder is smaller, as described in the work of P. Ulysse [27]. The formation of “onion rings” is unique to friction stir welding and related processes. Macroscopically, they are observed as a repetitive pattern on the transverse and lateral section of the weld and arise due to the cyclic change in grain size, second phase distribution and/or grain orientation. The patterns repeat at an interval equal to the linear distance travelled by the tool during each revolution. The origin of this pattern is still unsolved; research on this subject, presented, among others, by R.S. Mishra et al. [28,29], suggests that it is associated with the oscillation of the tool rotational axis about its traverse axis.

In order to determine the impact of FSP on the microstructure of the studied alloy, including the process parameters, statistical analysis of the SnSb and CuSn phases present in the microstructures was carried out. The results of the quantitative analysis of the cross-sectional area of the CuSn phase are presented in Figure 5; for the SnSb phase—in Figure 6. The obtained results confirm the refinement of the microstructure as a result of modification by FSP.

In the case of the cast material, CuSn precipitates with a size below 200 µm^2^ constituted about 50% of the studied population, and after FSP, 96–98% of the analyzed precipitates were included in this range. An increase in the rotational speed slightly influenced the refinement of the CuSn phase (Figure 5).

The obtained results of the size measurement of the SnSb phase precipitates also confirm their significant refinement after modification by FSP. It was found that with an increase in the rotational speed of the tool, there was an increase in the percentage of the smallest particles with a size under 2000 µm^2^. After the modification process with a rotational speed of 560 RPM, 80% of the studied population of SnSb precipitates was in this range. For comparison, the starting material contained in this range about 35% of the SnSb phase precipitates. The use of FSP modification practically eliminated precipitates larger than 12,000 µm^2^, whereas in the starting alloy these particles accounted for 22% of the studied population of precipitates.

The refinement of the SnSb and CuSn phase particles after the FSP modification process and the change in their distribution in the volume of the alloy only slightly influenced an increase in hardness (Figure 7).

The hardness of the starting alloy was 24 HB, and after FSP the hardness increased and amounted to 25 to 27 HB, respectively. The main factors affecting the increase in hardness may be the changes in the fraction and morphology of hard phases of the SnSb and CuSn compounds in the microstructure of the studied alloy after pin passage at different speeds.

### 3.2. Tribological Properties

Figure 8, Figure 9, Figure 10, Figure 11, Figure 12 and Figure 13 below show the results of tests of the tribological properties of the studied alloys, both in the initial state and also after various variants of FSP. Figure 8a,b presents curves showing changes in the coefficient of friction as a function of test time in technically dry friction conditions for both the applied loads of 50 and 100 N. Figure 8c,d presents the curves of changes in the coefficient of friction for the tribological test carried out in lubricated friction conditions using an oil dedicated for this type of material, with the trade designation TU 32. Analyses of changes in the coefficient of friction were also performed under conditions of the start–stop test (Figure 8e,f), which aims to simulate the actual conditions of intermittent operation of a turbine (multiple start-ups and stops). A detailed description of the start–stop process can be found in the work of Leszczyńska-Madej et al. [30].

Analysis of the course of changes in the coefficient of friction under conditions of technically dry friction (Figure 8a,b) shows that up to about 1000 s, i.e., along a distance of 250 m, the curves have a similar character; the coefficient of friction for all the investigated variants slightly increases almost linearly, but a significant spread of results was recorded. The lowest values, depending on the sliding distance, in this period, were gotten for the coefficient of friction for the starting material and the highest for the material after pin passage at 280 RPM. This is the time range in which run-in was achieved at tribological contact, and the value of the coefficient of friction is a function of the surface morphology, which depends on the size and fraction of individual precipitates on the surface of the studied element. After this period of stable friction course on the sliding distance after pin passage at speeds in the range of 280–450 RPM, there is a change in the character of the curves, resulting both from the increase in the coefficient of friction and its rapid changes over time. The curve for the starting material is stable until about 2800 s, while for the material after FSP modification (pin rotational speed of 560 RPM), it remains stable up to 2000 s, i.e., up to about 500 m. In these materials, there was also an increase in the coefficient of friction depending on time and there are numerous peaks related to their dynamic changes. This stable course of the coefficient of friction over the sliding distance from about 250 m to nearly 700 m, depending on the material, is advantageous from the point of view of the hypothetical failure of a turbine, e.g., due to a lack of lubricant for a period of time. Momentary dynamic changes in the coefficient of friction recorded on the curves in the form of steps/serrations result from the presence or disappearance of the adhesion phenomenon at the contact points of the rubbing surfaces. This phenomenon occurs in the contact of the tin-rich matrix with the matrix of the counter-specimen material (100Cr6 steel); therefore, its intensity depends strictly on the fraction and distribution of hard phases on the friction surface of the bearing alloy. A chemical interaction between the tin matrix and iron from the counter-specimen is also possible, which can also cause fluctuations in the coefficient of friction curve. Maps of the distribution of elements (Figure 9) on the surface after friction showed the transfer of iron to the bearing alloy.

The use of TU 32 oil dedicated for this type of material changes the character of the course of changes in the coefficient of friction (Figure 8c–f). The course of the curves along the sliding distance of 1000 m can be described as stable, with a gradual reduction in the coefficient of friction related to incubation of the oil in the frictional pair and the process of run-in of the friction surfaces. This corresponds to the second range of the variation of the friction force on the Stribeck curve, in which range the spreading of oil on the friction surfaces and the formation of a continuous insulating layer occur. The coefficient of friction decreases to a value below 0.1, which confirms the correct selection of oil (Figure 10). In both cases, the material has the lowest coefficient of friction after pin passage at 560 RPM, perhaps due to strong refinement of the hard phases, which are more evenly distributed in the matrix. They are also well fixed in the matrix and are not subjected to chipping, which has a positive impact on the course of friction, without increasing the value of the coefficient of friction. This situation is repeated on the curves of changes in the course of friction under the conditions of the start–stop test. The significant difference is in the peaks related to the stopping and restarting of the tester along the 1000 m sliding distance. At the moment of stopping, a certain amount of oil remains in the friction pair with wear products present in it. Then the oil is removed from the friction pair and there is a possibility of the wear products remaining between the frictional surfaces, increasing the friction force at the time of restarting. Since oil is supplied to the friction pair after a start-up and a layer is formed that insulates the surfaces, the coefficient of friction decreases quickly.

Significant fluctuations in the curves showing changes in the coefficient of friction during the tribological test are related to adhesive wear, which begins to play a significant role after the initial run-in. On observing the surface after friction (Figure 11a), areas formed as a result of breaking the bond formed during friction resulting from van der Waals forces are clearly visible. This mechanism, in addition to smearing of the matrix material, seems to be the main wear mechanism under conditions of technically dry friction. Inside the areas created as a result of smearing of the matrix on the bearing surface, there are visible cracks in the direction transverse to the direction of friction, which is related to the fatigue nature of wear in this type of friction system. The smearing of the matrix increases its fraction on the analyzed surfaces after friction while the pin rotational speed increases to 450 RPM. Such changes cause a significant increase in the coefficient of friction, especially at the lower load of 50 N (Figure 10a); a twofold increase in the load reduces the differences in the coefficient of friction between the studied variants of pin rotation, but the dependence remains the same (Figure 10b). The map of the distribution of elements on the analyzed surfaces after friction also shows the presence of oxides that can be identified as tin and iron oxides, which may come from the counter-specimen—confirmed by the aforementioned adhesive wear and chemical interaction between tin and iron (it is also the probable cause of the fluctuations of the coefficient of friction curve). In the studied system, these oxides are most likely to be obtained, which results from analysis of the Ellingham–Richardson diagram. The oxides, especially iron and tin, present on the friction surface, do not form a permanent bond with the substrate and are easily detached from the surface and may participate in the friction. However, the properties of these oxides have little effect on the course of friction and are quickly removed from the friction pair by the rotating counter-specimen.

There were significant differences in the surface morphology after friction in the starting material and in the FSP materials (Figure 11a). The smearing of the tin-rich matrix in the post-FSP materials was more even, which was mainly due to the refinement of the SnSb phase. Between these areas of tin smear, significant chipping occurred in the starting material due to intensive adhesive wear, which was not found in the post-FSP materials. Initiating adhesive wear leads to a significant increase in the coefficient of friction.

The nature of the wear of the bearing alloys results from the interaction of the soft tin matrix and harder precipitates of the CuSn and SnSb phases. The use of TU 32 oil completely changes the course of friction and the wear mechanisms occurring in the friction pair in comparison to dry friction, regardless of the applied load. The clearly decreasing coefficient of friction (Figure 10a,b) and the surface morphology after friction (Figure 11b,c and Figure 12) make it possible to unequivocally conclude that adhesive wear, dangerous in this type of system, can be completely eliminated. The curves of changes in the coefficient of friction usually have a stable course and it is reflected by the observed surfaces after friction. The small fluctuations present on them resulted from the chipping of hard phases from the matrix, which is illustrated in Figure 11 and Figure 12. The largest fluctuations are visible on the curve for the material after modification with the pin rotating at 450 RPM, and confirmation of this fact is the numerous chippings present in the observed surface area after friction; therefore, these particles for some time participated in the friction as an abrasive. The chipped off particles act in the friction pair for a very limited time because only individual scratches are visible on the surfaces after friction, which follow the direction of friction. The scratches run both through the matrix and the hard particle phases. On analyzing the surface morphology after friction in relation to changes in the coefficient of friction, it can be concluded that the abrasive wear occurring in the initial period of friction, as the dominant mechanism, has a positive effect on the coefficient of friction of the starting material; it is possible due to the greater effective distances between the hard phase particles in the tin-rich matrix. Thus, the chromium carbides present in the steel counter-sample on the surface have the possibility of penetrating into the tin matrix or smeared over the tin surface more than in the samples with greatly refined particles of CuSn and SnSb phases, which make scratching difficult. This mechanism is also present in the test with lubrication at a higher load in both the continuous friction and start–stop types. An increased load reduces the distance between the rubbing surfaces, which may result in mutual contact of both the surfaces in places where surface irregularities occur (e.g., carbides in the counter-sample), where it is easier to scratch and groove; the coefficient of friction will be lower, but the presence of abrasive wear in the presence of the lubricant results in scuffing with longer operating times of this type of friction pairs.

Increasing the load results in faster run-in and therefore faster achievement of stabilized friction conditions, especially in an alloy with the correct microstructure (starting alloy). Nevertheless, increasing the importance of adhesion and its course cause the coefficient of friction to equalize after about 3400 s (Figure 8b). Unfortunately, this favorable course of changes in the coefficient of friction in the initial stage does not translate into the weight loss (Figure 13), which for the starting material is the largest—mainly due to the action of adhesion and the breaking of much larger fragments of the material than in the case of the materials with grain refinement due to FSP.

In laboratory conditions, the limited capacity of the oil pan during the friction process causes chipped off particles to be present in the oil used and they form a suspension with their increasing concentration. In the studied range of sliding distance, the above-discussed phenomenon does not affect the friction course as the instantaneous value of the coefficient of friction decreased continuously. Analysis of the curves of changes in the coefficient of friction during the start–stop test (Figure 8e,f) also shows that for the applied load of 50 N, its values slightly decrease along the entire sliding distance, while increasing the load to 100 N causes its continuous reduction, but its average value is higher than for the 50 N load. It is probable that for both the applied loads the whole test takes place at the run-in stage, wherein increasing the pressure causes its faster course.

Studies conducted in other centers [19,20] have also shown the possibility of reducing the coefficient of friction as a result of applying appropriate surface treatment. Depending on the laser parameters, laser remelting of the surface layer allows the value of the coefficient of friction under lubricated friction conditions to be reduced to the range of 0.06–0.12. However, at very low loads of 5–15 N, these values are higher than for materials after FSP tested under the load of 50 N [19]. The coefficient of friction results obtained by surface texturing, where the coefficient of friction oscillates around the value of 0.06, were similar to the results obtained by FSP [21]. In both cases, analysis of the wear mechanisms was also carried out, and as a result of laser remelting of the surface, delamination and numerous chipping particles were additionally found.

In order to compare the tribological properties of the studied materials, the weight losses during the test were also determined (Figure 13). Weighing of the samples after the lubricated tests was performed after removing the oil film from their surface.

The determined weight losses for the test carried out in technically dry friction conditions show that increasing the load causes its significant increase. The change in load is also associated with a change in the nature of the impact of FSP—in particular, the change in the speed of the pin on the weight loss. The samples from the starting material during the test in technically dry friction conditions are characterized by the lowest weight loss in relation to the modified samples, whereas increasing the load to 100 N causes the opposite effect—the use of FSP improves the wear resistance characterized by the weight loss. For the load of 100 N in technically dry friction conditions, the weight loss of the samples after FSP is similar and falls within the range of 1.18–1.24%, and application of the 50 N load causes a decrease in wear with increasing rotational speed. The factor determining these changes is the refinement of the microstructure.

Using TU 32 oil improves the wear resistance of materials after FSP, both those after friction for a 1000 m distance and those after the start–stop test compared to dry friction carried out under the same conditions. Under the load of 50 N, the weight losses were very low, both for the 1000 m distance, where basically the run-in stage was observed all the time, and for the 10,000 m distance with stopping every 1000 m; the use of FSP reduced the weight loss. Increasing the load to 100 N gives a similar effect in terms of changes in the amount of wear, but the weight loss value clearly increases. This may be due to the characteristics of the oil used, which is dedicated for work at lower loads. Increasing the rotational speed of the pin under lubricated friction conditions significantly improves the wear resistance of this alloy, especially when using the rotational speed of 560 RPM, where the greatest refinement of the phases present in the studied alloy was observed.

## 4. Conclusions

The conducted tests allowed the following conclusions to be formed:The use of friction stir processing for surface treatment of the SnSbCu bearing alloy causes significant changes in the morphology of the hard phases, including their refinement—confirmed by the increased fraction of the finest SnSb particles with areas under 2000 µm^2^ and CuSn with areas under 200 µm^2^. In addition, using modification with the highest rotational speed of the tool—560 RPM—results in the greatest refinement of the microstructure.The use of FSP reduced the weight loss of the studied materials compared to the base alloy. The wear resistance increased when increasing the pin rotational speed, which is closely related to the refinement of the hard phases and their more even distribution in the matrix after modification at the speed of 460 RPM, and especially at 560 RPM—when the weight loss was about three times lower than for the starting material, while increasing the hardness from 24 to 27 HB.The refinement of the CuSn and SnSb phases obtained in FSP at the pin speed of 560 RPM has very positive effects on the hardness and tribological properties of the tested alloy, which may indicate the possibility of longer failure-free operation under lubricated friction conditions, postponing the effect associated with chipping wear (pitting).

## Figures and Tables

**Figure 1 materials-13-05826-f001:**
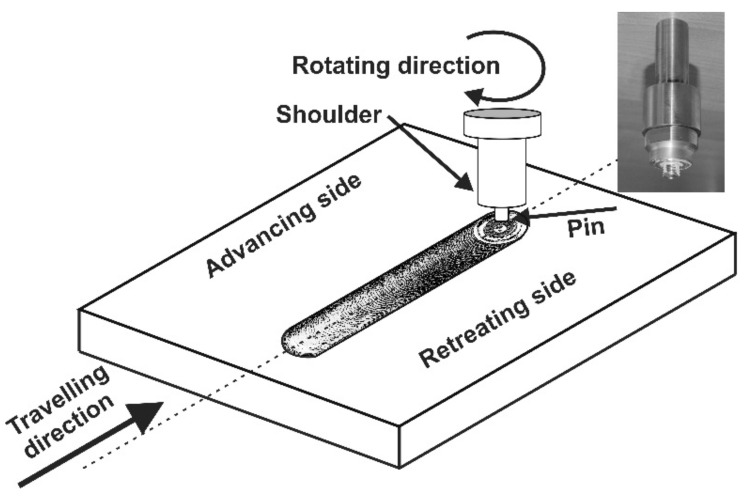
Schematic diagram of friction stir processing (FSP) modification; this figure also shows a photo of the Triflute tool.

**Figure 2 materials-13-05826-f002:**
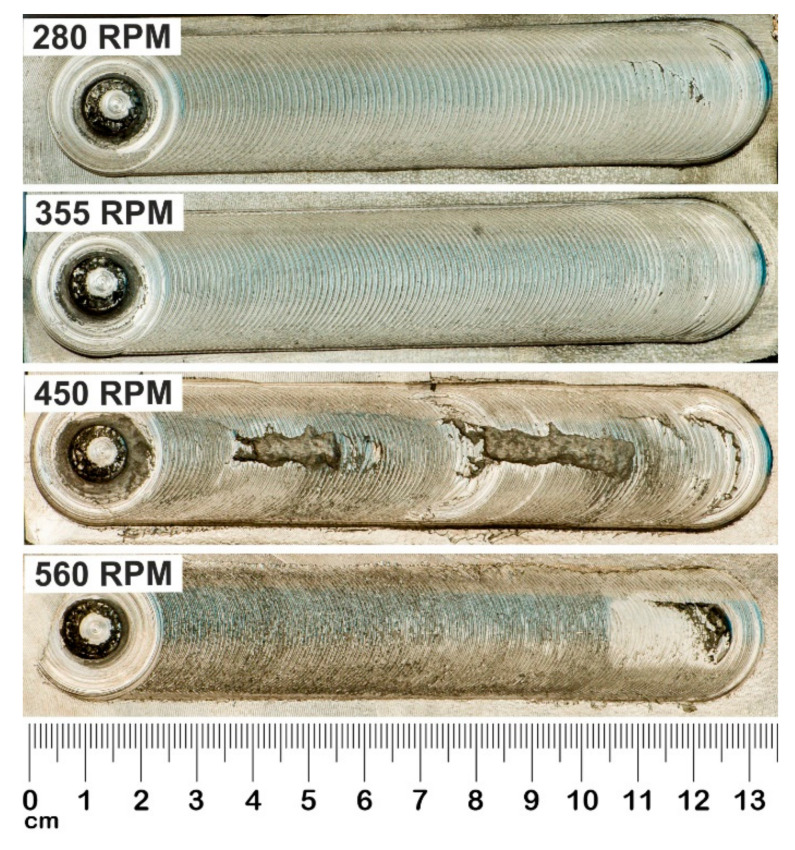
Photo of samples after FSP modification process.

**Figure 3 materials-13-05826-f003:**
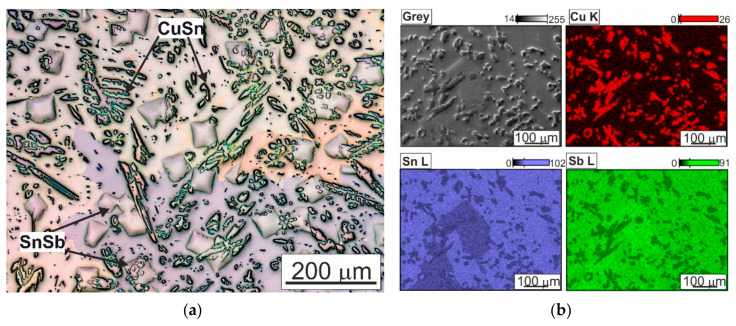
Microstructure of SnSb11Cu6 alloy in initial state after casting; (**a**) photo taken using a light microscope; (**b**) microstructure and maps of distribution of elements: Cu, Sn, Sb; SEM.

**Figure 4 materials-13-05826-f004:**
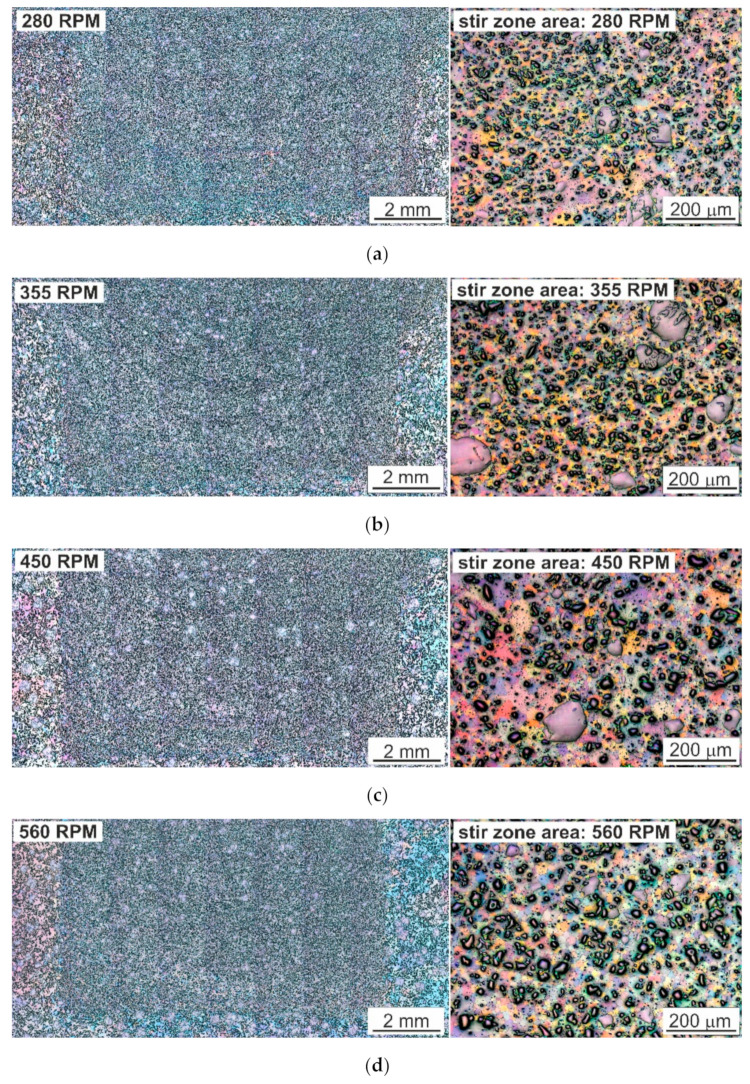
Microstructure of SnSb11Cu6 alloy after FSP with visible unmodified area (left side); photo on the right shows the modified area. (**a**) 280 RPM, (**b**) 355 RPM, (**c**) 450 RPM, (**d**) 560 RPM; LM.

**Figure 5 materials-13-05826-f005:**
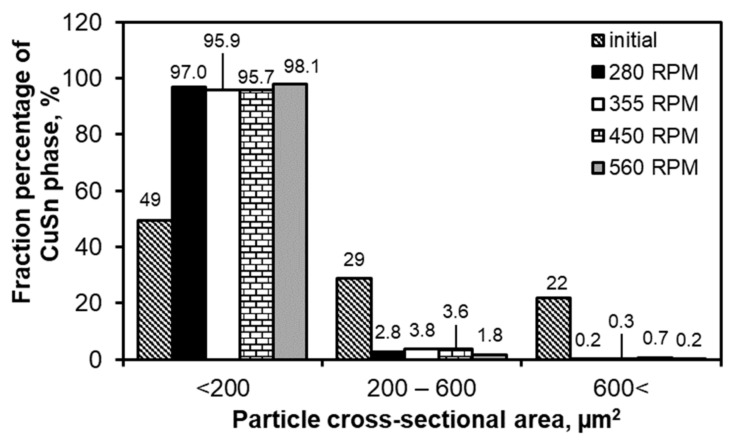
Results of statistical analysis of CuSn precipitates depending on FSP conditions; results are related to the alloy in initial state after casting.

**Figure 6 materials-13-05826-f006:**
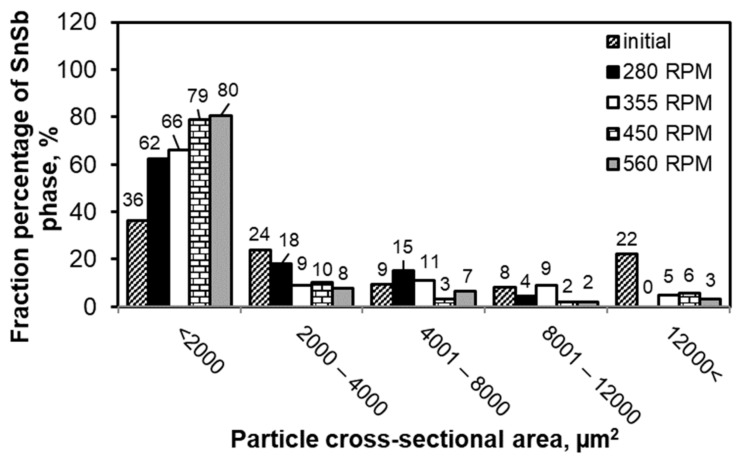
Results of statistical analysis of SnSb phase depending on FSP conditions; results are related to alloy in initial state after casting.

**Figure 7 materials-13-05826-f007:**
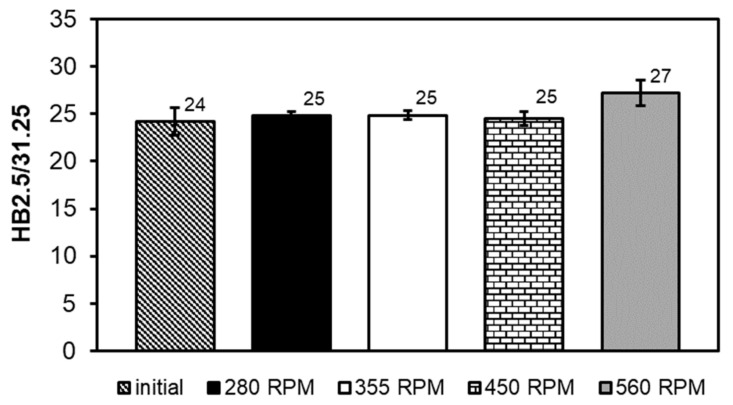
Comparison of hardness of SnSb11Cu6 alloy before and after FSP.

**Figure 8 materials-13-05826-f008:**
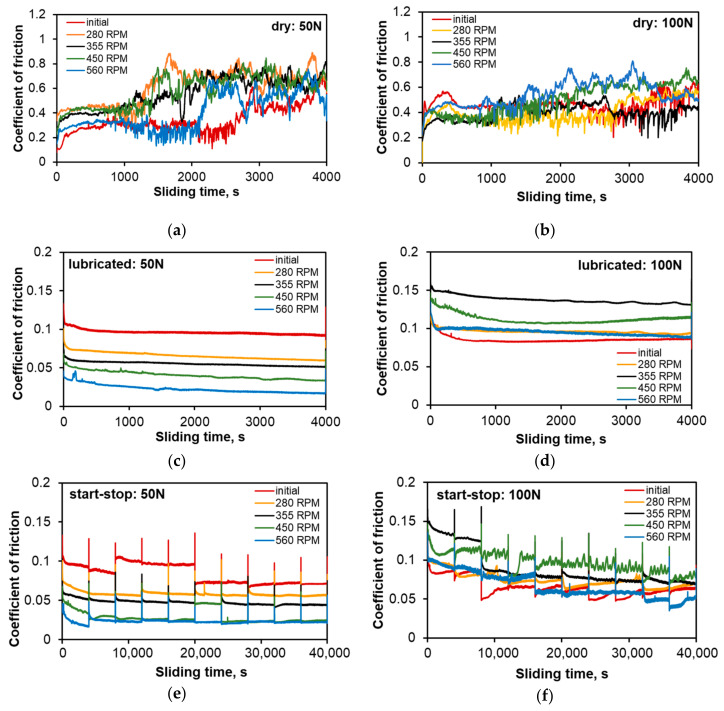
Coefficient of friction of SnSb11Cu6 alloy before and after FSP as a function of time and process conditions; (**a**) technically dry friction, load 50 N; (**b**) technically dry friction, load 100 N; (**c**) lubricated friction, load 50 N; (**d**) lubricated friction, load 100 N; (**e**) lubricated friction—start–stop, load 50 N; (**f**) lubricated friction—start–stop, load 100 N.

**Figure 9 materials-13-05826-f009:**
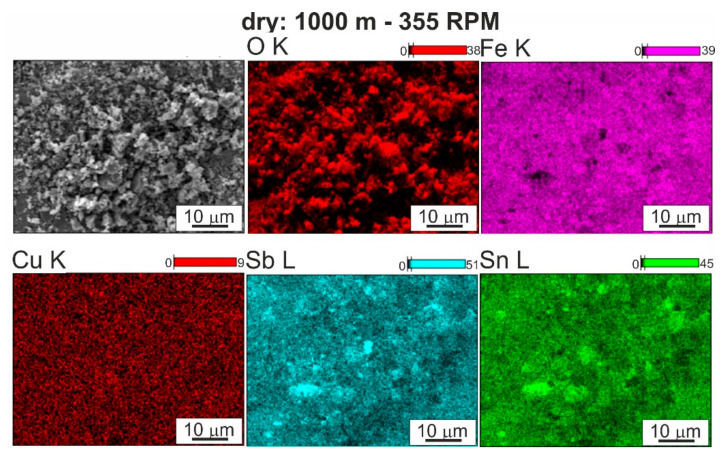
Surface after friction of SnSb11Cu6 alloy FSP modified using speed 355 RPM and map of distribution of elements: O, Fe, Cu, Sn, Sb; technically dry friction; SEM.

**Figure 10 materials-13-05826-f010:**
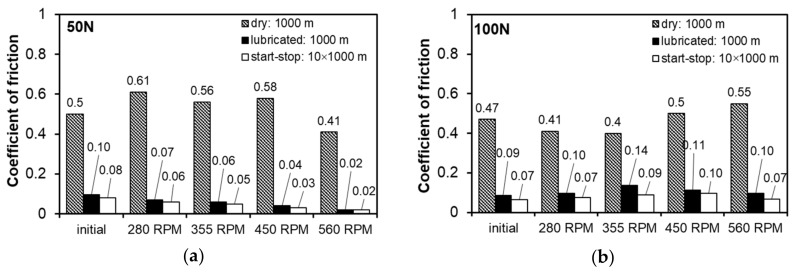
Coefficient of friction: (**a**) determined at load of 50 N, (**b**) determined at load of 100 N.

**Figure 11 materials-13-05826-f011:**
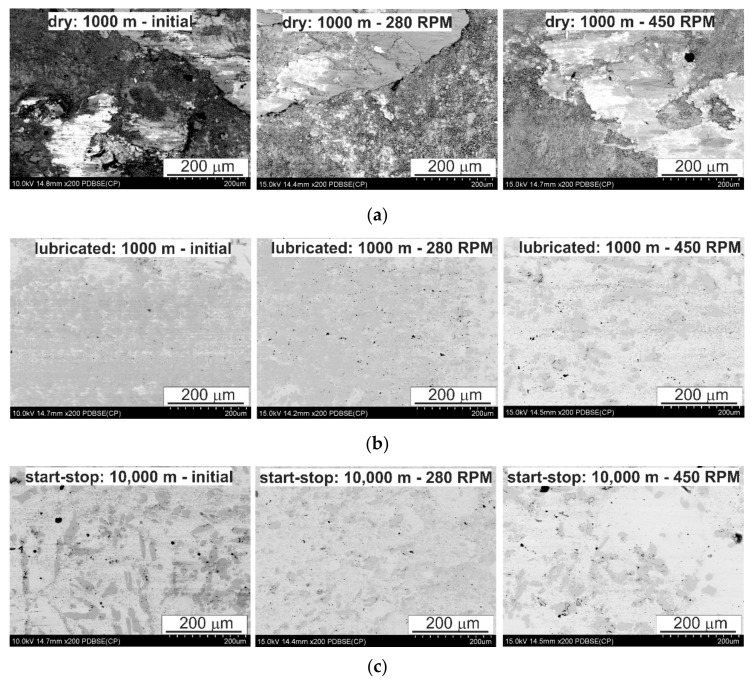
Characteristic sample surface of SnSb11Cu6 alloy after tribological tests in different conditions; (**a**) dry friction: 1000 m, (**b**) lubricated friction: 1000 m, (**c**) start-stop test: 10,000 m; SEM.

**Figure 12 materials-13-05826-f012:**
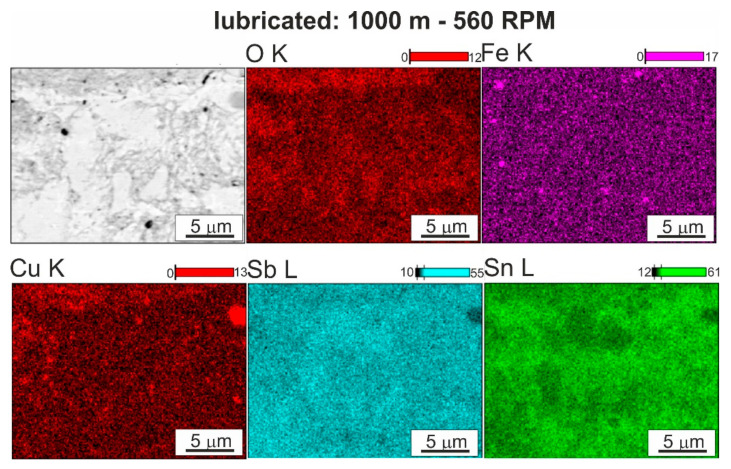
Surface after friction of SnSb11Cu6 alloy FSP modified using speed of 560 RPM and maps of distribution of elements: O, Fe, Cu, Sn, Sb; lubricated friction; SEM.

**Figure 13 materials-13-05826-f013:**
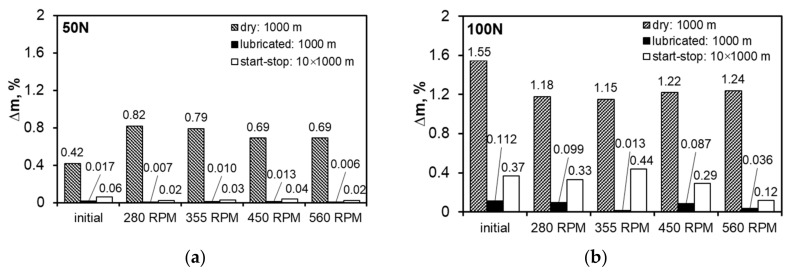
Weight loss: (**a**) determined at load of 50 N, (**b**) determined at load of 100 N.

**Table 1 materials-13-05826-t001:** Chemical composition of investigated alloy, wt%.

Name of Alloy		Chemical Composition, wt%			
Grade mark	designation	Sn	Pb	Sb	Cu
SnSb11Cu6	B83	rest	0.18	10.52	6.06

**Table 2 materials-13-05826-t002:** Wear test parameters.

Sliding Contact	Counter-Specimen	Rotational Speed	Load	Sliding Distance
Technically dry *	φ 49.5 mm, steel 100Cr6, heat-treated, with hardness of 55 HRC	163 RPM	50 N	1000 m
Lubricated				1000 m
Lubricated				10 cycles × 1000 m
Technically dry *			100 N	1000 m
Lubricated				1000 m
Lubricated				10 cycles × 1000 m

* Technically dry: the friction process was carried out dry without lubricants at low air humidity (approximately 30%) at ambient temperature.

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
