# Peer review of "Effects of the Processing Parameters of Friction Stir Processing on the Microstructure, Hardness and Tribological Properties of SnSbCu Bearing Alloy"

_materials, 2020, doi:10.3390/ma13245826_

Round 1
Reviewer 1 Report
The authors investigated the Effect of processing parameters of friction stir processing on microstructure, hardness and tribological properties of SnSbCu bearing alloy. The topic is interesting and the content is important. However, the manuscript is not well prepared and revision is required before the publication.
Though the authors got the effects of the parameters on the performance at a definite range, the optimized parameter range hasn’t been suggested. The authors should have discussions about the potential parameter range to get the optimized performance.
Though the improvement in wear resistance compared to that of the base alloy is reported, could the authors compare the properties of the alloy processed by FSW with those processed by the other processing methods, e.g., laser remelting, surface texturing, and so on. Though FSW could lead to the increase in performance, it could also improve the production cost.
The conclusion section should be simplified. Only the really new findings could appear in the conclusion section.
The following suggestions also need to be revised.
- All the figures provided in the manuscript have a very low resolution.
- 11, the scale bar is not clear.
- 2, scale bar is required.
- The manuscript should be thoroughly edited to reach the requirement of publication.
Author Response
Dear Reviewer,
We greatly appreciate your thoughtful remarks that helped improve the manuscript. In the following, we give a point-by-point reply to your remarks. The changes and amendments have been introduced to the text of the publication. All the changes have been highlighted using “Track Changes”.
Remark #1
The authors investigated the Effect of processing parameters of friction stir processing on microstructure, hardness and tribological properties of SnSbCu bearing alloy. The topic is interesting and the content is important. However, the manuscript is not well prepared and revision is required before the publication. Though the authors got the effects of the parameters on the performance at a definite range, the optimized parameter range hasn’t been suggested. The authors should have discussions about the potential parameter range to get the optimized performance.
Response:
Based on the obtained results, it can be concluded that the pin speed of 560 RPM is optimal from the point of view of the obtained properties.
The refinement of the CuSn and SnSb phases obtained in FSP at the pin speed of 560 RPM has a very positive effect on the hardness and tribological properties of the tested alloy, which may indicate the possibility of longer failure-free operation under lubricated friction conditions, postponing the effect associated with chipping wear (pitting).
Remark #2
Though the improvement in wear resistance compared to that of the base alloy is reported, could the authors compare the properties of the alloy processed by FSW with those processed by the other processing methods, e.g., laser remelting, surface texturing, and so on. Though FSW could lead to the increase in performance, it could also improve the production cost.
Response:
Unfortunately, the methods of testing the tribological properties (mainly ball-disc) used in the articles on laser Babbit surface treatment exclude the comparison of weight loss, but the coefficients of friction and wear mechanisms can be compared, which was incorporated in the article.
Studies conducted in other centers [19, 20] have also shown the possibility of reducing the coefficient of friction as a result of applying appropriate surface treatment. Depending on the laser parameters, laser remelting of the surface layer allows the value of the coefficient of friction under lubricated friction conditions to be reduced to the range of 0.06 - 0.12. However, at very low loads of 5 - 15 N, these values are higher than for materials after FSP tested under the load of 50 N [19]. The coefficient of friction results obtained by surface texturing, where the coefficient of friction oscillates around the value of 0.06, were similar to the results obtained by FSP [21]. In both cases, analysis of the wear mechanisms was also carried out, and as a result of laser remelting of the surface, delamination and numerous chipping particles were additionally found.
References:
- Ni Y.; Dong G.; Tong Z.; Li X.; Wang W.: Effect of laser remelting on tribological properties of Babbitt alloy. Materials Research Express 2019, 6, 1-10.
- Potekhin B.A.; Il’yushin V. V.; Khristolyubov A. S. Effect of casting methods on the structure and properties of tin babbit. Metal Science and Heat Treatment 2009, 51/7-8, 378-382.
- Zhang H.; Zhang D.Y.; Hua M.; Dong G.N. A study on the tribological behaviour of surface texturing an Babbit alloy under mixed or starved lubrication. Letter 2014, vol. 56, 305-315.
Remark #3
The conclusion section should be simplified. Only the really new findings could appear in the conclusion section.
Response:
The conclusions have been redrafted as suggested.
Remark #4
The following suggestions also need to be revised.
- All the figures provided in the manuscript have a very low resolution.
- 11, the scale bar is not clear.
- 2, scale bar is required.
- The manuscript should be thoroughly edited to reach the requirement of publication.
Response:
The figures and article have been revised as suggested.

Reviewer 2 Report
Dear authors,
The paper is well organized and suitable for publication in materials. However, some minor revisions have to be done before publishing:
The authors conclude that the fine distribution of the particles after fsp is beneficial for the wear behavior. This is true for many of their results. However, there are some point, which cannot explained by this:
- Why does the initial samples have the latest increase in coefficient of friction at the dry 50 N test?
- Why are the 280, 355 and 450 rpm so much worse compared to the 560 rpm sample, although the microstructure is refined similarly. The behavior under dry conditions, wet 100 and start-stop 100 N is even worse than that of the initial samples.
- The initial sample has the highest coefficient of friction at 50 N but the lowest value at 100 N in the wet test. Please explain this behavior more detailed.
Author Response
Dear Reviewer,
We greatly appreciate your thoughtful remarks that helped improve the manuscript. In the following, we give a point-by-point reply to your remarks. The changes and amendments have been introduced to the text of the publication. All the changes have been highlighted using “Track Changes”.
Additionally, linguistic correction was performed once again.
Remark #1
Why does the initial samples have the latest increase in coefficient of friction at the dry 50 N test?
Response
The coefficient of friction determined under the conditions of technically dry friction with the load of 50 N increases due to the appearance of adhesive wear. This type of wear occurs later than in the material after FSP treatment, but occurs with greater intensity.
There are significant differences in the surface morphology after friction in the starting material and in the FSP materials (Fig. 11a). The smearing of the tin-rich matrix in the post-FSP materials is more even, which is mainly due to the refinement of the SnSb phase. Between these areas of tin smear, significant chipping occurs in the starting material due to intensive adhesive wear, which was not found in the post-FSP materials. Initiating adhesive wear leads to a significant increase in the coefficient of friction.
Remark #2
Why are the 280, 355 and 450 rpm so much worse compared to the 560 rpm sample, although the microstructure is refined similarly. The behavior under dry conditions, wet 100 and start-stop 100 N is even worse than that of the initial samples.
Response
Considering the course of changes only in the coefficient of friction, one could get such an impression and the test method can be easily excluded. However, by summing up the properties such as the average coefficient of friction, weight loss and the existing friction mechanisms, it can be concluded that the use of FSP is beneficial for bearing alloys. Unfortunately, it is difficult to unequivocally indicate the relationship between grain refinement and the obtained coefficient of friction and weight loss for all the variants, and to indicate unequivocally that increasing the speed causes continuous improvement of the properties. Nonetheless, the refinement of the CuSn and SnSb phases obtained in the FSP process at the pin speed of 560 RPM has a very positive effect on the tribological properties of the tested alloy and may indicate the possibility of longer failure-free operation under lubricated friction conditions, postponing the effect associated with chipping wear (pitting).
On analyzing the surface morphology after friction in relation to changes in the coefficient of friction, it can be concluded that the abrasive wear occurring in the initial period of friction, as the dominant mechanism, has a positive effect on the coefficient of friction of the starting material; it is possible due to the greater effective distances between the hard phase particles in the tin-rich matrix. Thus, the chromium carbides present in the steel counter-sample on the surface have the possibility of penetrating into the tin matrix or smeared over the tin surface more than in the samples with greatly refined particles of CuSn and SnSb phases, which make scratching difficult. This mechanism is also present in the test with lubrication at a higher load in both continuous friction and start-stop types. An increased load reduces the distance between rubbing surfaces, which may result in mutual contact of both surfaces in places where surface irregularities occur (e.g. carbides in the counter-sample), where it is easier to scratch and groove; the coefficient of friction will be lower, but the presence of abrasive wear in the presence of the lubricant results in scuffing with longer operating times of this type of friction nodes.
Remark #3
The initial sample has the highest coefficient of friction at 50 N but the lowest value at 100 N in the wet test. Please explain this behavior more detailed.
Response
Increasing the load results in faster run-in and therefore faster achievement of stabilized friction conditions, especially in an alloy with the correct microstructure (starting alloy). Nevertheless, increasing the importance of adhesion and its course cause the coefficient of friction to equalize after about 3400 s (Fig. 8 b). Unfortunately, this favorable course of changes in the coefficient of friction in the initial stage does not translate into the weight loss (Fig.13), which for the starting material is the largest - mainly due to the action of adhesion and breaking of much larger fragments of the material than in the case of the materials with grain refinement due to FSP.
Tin-based bearing alloys are not intended for use in dry friction conditions. In addition to testing as a reference for lubricated friction, the purpose of the dry friction test is to determine the behavior of the alloy after lubrication failure and cut-off of the grease/oil supply.

Reviewer 3 Report
For being acceptable authors should include in the abstract the novelty of the study in particular pratical implications.
In introduction a RQ is necesary to guide the reader through the all paper.
From line 45-56 references are needed. The work referenced implies all this chapter?
Author Response
Dear Reviewer,
We greatly appreciate your thoughtful remarks that helped improve the manuscript. In the following, we give a point-by-point reply to your remarks. The changes and amendments have been introduced to the text of the publication. All the changes have been highlighted using “Track Changes”.
Remark #1
For being acceptable authors should include in the abstract the novelty of the study in particular practical implications
Response
The abstract has been supplemented and the novelty of this type of method has been highlighted in relation to bearing alloys.
Remark #2
In introduction a RQ is necesary to guide the reader through the all paper.
Response
The introduction was supplemented with the aim of the work.
Remark #3
From line 45-56 references are needed. The work referenced implies all this chapter?
Response
Thank you very much for your observation; most likely when the text was formatted the references were deleted by mistake. This has been corrected.

Round 2
Reviewer 1 Report
Could be accepted